# Geospatial analysis of the associations between environmental contamination with livestock feces and children with chronic fascioliasis in the Anta province of Cusco, Peru

Melinda Barbara Tanabe[1], John Prochaska[2], Maria Luisa Morales[3,4], Martha Lopez[3,4], Benicia Baca-Turpo[3,4], Eulogia Arque[3,4], Miguel Mauricio Cabada [1,3,4] *

1 Division of Infectious Diseases, Department of Medicine, University of Texas Medical Branch, Galveston, Texas, United States of America, 2 Graduate School of Biomedical Sciences, University of Texas Medical Branch, Galveston, Texas, United States of America, 3 Alexander von Humboldt Tropical Medicine Institute, Department of Medicine, Universidad Peruana Cayetano Heredia, San Martin de Porres, Lima, Perú, 4 UPCH–UTMB Collaborative Research Center—Cusco, Universidad Peruana Cayetano Heredia, Wanchaq, Cusco, Peru

* micabada@utmb.edu

**Data Availability Statement:** The raw data used for this manuscript is available at the Harvard

## Abstract

*Fasciola hepatica* is a neglected parasitic infection with significant human health and live-stock industry impact. The Andean Altiplano harbors an estimated 50% of the *Fasciola's* world infection burden. There is scarce data regarding the spatial associations between different *Fasciola* hosts. In this project, we aimed to determine the geospatial relationships between *Fasciola* eggs passed in feces of different livestock species and the risk of infection among each household as a unit. We used data from a cross-sectional study evaluating children and livestock feces for *Fasciola* infection around households in three districts of Anta province, in the Cusco region of Peru. Each sample was geographically tagged and evaluated for fascioliasis using microscopy methods. A total of 2070 households were included, the median age was 9.1 years (6.7–11.8), 49.5% were female, and 7.2% of the households had at least one infected child. A total of 2420 livestock feces samples were evaluated. The infection rate in livestock samples was 30.9%. The highest infection rate was found in sheep with 40.8%, followed by cattle (33.8%), and swine (26.4%). The median distance between a household with an infected child to a positive animal sample was 44.6 meters (IQR 14.7–112.8) and the distance between a household with no infected children to a positive animal sample was 62.2 meters (IQR 18.3–158.6) (p = 0.025). The multivariable logistic regression adjusted by presence of poor sanitation, unsafe water consumption, altitude, and presence of multiple infected children per household demonstrated an association between household infection and any cattle feces at a 50 meters radius (Uninfected: OR 1.42 (95%CI 1.07–1.89), p = 0.017. Infected: OR 1.89 (95%CI 1.31–2.73), p = 0.001), positive cattle feces at a 100 meters radius (OR 1.35 (95% CI 1.08–1.69), p = 0.008), and negative cattle feces at a 200 meters radius (OR 1.08 (95% CI 1.01–1.15), p = 0.022). We identified potential hot and cold spots for fascioliasis in the Anta province. An association between environmental contamination with feces from different livestock species and infected children in rural

Dataverse public repository (https://dataverse.
harvard.edu). Full data cannot be shared publicly as
it violates human subject research compliance as it
provides identifiable information, therefore all data
but geographic location coordinates are available.
The database without identifiable information can
be requested via this repository. https://doi.org/10.
7910/DVN/YHEQXU.

**Funding:** M.M.C. work was supported by the
National Institute for Allergy and Infectious
Diseases, National Institutes of Health grant no.
1R01AI104820 (https://www.niaid.nih.gov). The
funders had no role in study design, data collection
and analysis, decision to publish, or preparation of
the manuscript.

**Competing interests:** The authors have declared
that no competing interests exist.

households was found in our study. Local health authorities may apply this strategy to esti-
mate the risk of infection in human populations and apply targeted interventions.

## Author summary

*Fasciola hepatica* is the parasite that causes fascioliasis in the Andes region. The factors
associated with transmission of the disease among humans have not been well described
in the area studied. In this study, we evaluate the spatial relationships between infected
households and the environmental burden of *Fasciola* eggs passed by different livestock in
the Cusco region of Peru. We geographically marked and tested for fascioliasis feces pro-
vided by children and livestock. Infected and uninfected cattle as well as sheep feces were
more likely to be in proximity to human households with infected children than to house-
holds without infected children. We were able to identify and map the areas of higher and
lower infections rates. This information and strategy may be used in the control of the
disease.

## Introduction

*Fasciola hepatica* is the causative agent of fascioliasis in the Andes region. It is considered an emer-
gent infectious disease, with reports suggesting the establishment of new endemic areas in locations
where sporadic transmission occurred in the past [1–3]. Climate change and man-made environ-
mental modifications are probably associated with the increased prevalence and geographic distribu-
tion of Fasciola [4]. Fascioliasis causes significant financial burden in the livestock industry around
the world. Over 90% of the human burden is located in low-resource areas in small farming com-
munities [5,6]. School age children are disproportionally vulnerable to the infection [7,8].

   *Fasciola hepatica* has a complex life cycle that includes snail intermediate hosts and a variety
of mammalian definite hosts (Fig 1). Animals and humans with fascioliasis often have overlap-
ping endemic areas. A Venezuelan study reported that animal and human fascioliasis was asso-
ciated with snail presence and consumption of untreated water [9]. Mas-Coma et al. reported
a prevalence of fascioliasis between 0–65% in cattle and 0–68% in humans with geographic
overlap in different host endemic areas in the Bolivian Altiplano [10–11]. In Brazil, the low
report of fascioliasis in humans over 60 years differed from the high reports of animal infection
in the Southern states, specifically cattle [12–13]. In part, this could be explained by lack of sur-
veillance and diagnosis in humans, as the disease seems to be spread in an area with overlap-
ping human and livestock disease, however epidemiology is far from being elucidated [12].
The inter-host dynamics in *Fasciola* infection are not fully understood and may explain the
inconsistencies between studies as well as differences in transmission patterns. Although no
direct fecal-oral transmission of fascioliasis is possible, the role of *Fasciola's* environmental
load caused by infected livestock in human transmission needs further scrutiny.

   In this study, we aimed to determine the geospatial association between *Fasciola* eggs
passed in feces of different livestock species and the risk of infection among each household in
the Cusco region of Peru.

## Methods

### Ethics statement

The Institutional Ethics Committee of Universidad Peruana Cayetano Heredia and the Uni-
versity of Texas Medical Branch Institutional Review Boards approved the study protocol. The

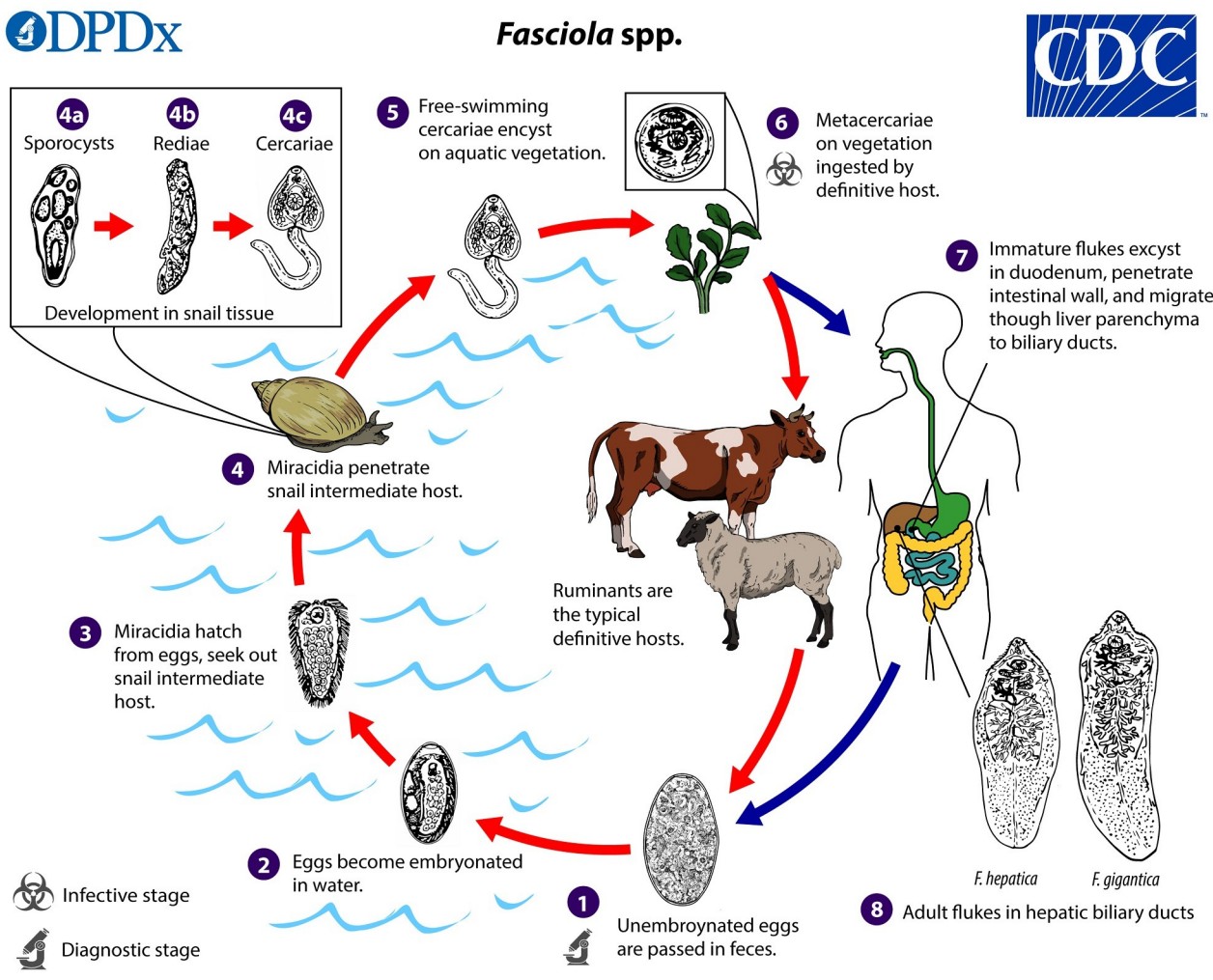

**Fig 1. Lifecycle of *Fasciola hepatica*, reproduced with permission from DpDx CDC *Fasciola spp.* lifecycle [14].**

consent process was performed in Quechua or Spanish language according to subject's preference. A written and verbal informed consent was obtained from the children's guardians and assent was obtained from children older than 6 years before any study procedure. Children with fascioliasis were referred to the Ministry of Health for treatment with triclabendazole.

We performed a cross-sectional study among children in 26 communities of the Anta Province in the Cusco region of Peru [15]. Children between 3 and 16 years of age enrolled in pre-schools and schools of the districts of Ancahuasi, Anta, and Zurite, were invited to participate between August of 2013 and August of 2018. Containers and instructions to collect only their own feces sample were provided to each child at pre-school and schools. Freshly produced feces samples were collected the next morning in consecutive days. Children that failed to produce a feces sample were given instructions again to collect a sample the next day. Each child's household was visited within 2 weeks of enrolment. Field workers visited each household to collect demographic, socio-economic, and epidemiological data. The Simple Poverty Score-card (Microfinance Risk Management LLC) was used to evaluate the probability of the child's household to be under the poverty lines [16]. The entrance to the main living space of the house was geographically tagged at the time of visit. For the present study, we selected all

children that provided at least one fecal sample for testing and had Global Position System (GPS) coordinates for the house where they resided regardless of infection status. Households were considered the analysis unit and houses with one or multiple children infected were defined as *Fasciola* positive households. The analysis was adjusted for the presence of multiple infected children in one household. If no infected children were found in a house, the household was defined as *Fasciola* negative household. The field workers collected up to four fresh fecal samples passed by different livestock species in the immediate vicinity of each house. Livestock samples were considered fresh if they were still moist at the time of collection. The livestock samples closest to the household were selected, geographically tagged, and classified according to the species. For the present study we focused on feces passed by cattle, swine, and sheep and selected one sample per species in each household. We were unable to ascertain if feces samples were from livestock owned by a specific household, but the proximity between the living space and the samples made ownership likely. If no GPS data was available for a livestock sample, the house coordinates were used as a proxy only for the radius analysis but not for distance analysis.

All fecal samples were collected in individual clean containers, maintained at 4–8˚C, and transported as soon as possible to the laboratory at the Cusco Branch–Tropical Medicine Institute of Universidad Peruana Cayetano Heredia in Cusco city. Children's fecal samples were preserved fresh and in 10% formalin. Fresh samples were tested by the Kato–Katz method within 24–48 hours of collection. The samples preserved in 10% formalin were tested by the Lumbreras rapid sedimentation method within a week. [17] Chronic fascioliasis in children was defined as having *F. hepatica* eggs in any of the samples collected. Livestock feces were analyzed by the Lumbreras rapid sedimentation method for the presence of *Fasciola* eggs. Internal and external quality control procedures for data collection, data management, and laboratory procedures were used in the study. Known positive and negative stool samples were introduced in the laboratory routine to evaluate technician proficiency. Positive microscopy results were confirmed by a second observer.

The statistical analysis was performed using the SAS Statistical Package v. 9.4 (SAS Institute Inc., 2013). We used ArcGIS v. 10.7.1 (ESRI, 2019) for the analysis of infection distribution and inter-host associations. We calculated frequencies, means ± standard deviation (±SD), and medians with interquartile ranges (IQR) to describe the distribution of the variables. We calculated the median distances from the different households to the closest infected or non-infected livestock feces. Livestock feces without GPS data were excluded from the radius analysis. The student-*t* test, Fisher/$X^2$ test, or Mann-Whitney-U test were used accordingly to compare the households with and without *Fasciola* infection. A backwards logistic regression analysis was performed using *Fasciola* positive household as the dependent variable to evaluate the spatial association between infection status of feces samples from different livestock species. We delineated areas around each household at 50, 100, and 200-meter radius and counted the number of positive and negative livestock feces samples within each radius area. In each radius model, we included the covariates animal feces count inside each radius area per species and infection status (sheep positive, sheep negative, cattle positive, cattle negative, pig positive, and pig negative). All models were adjusted by reported poor sanitation, unsafe water intake, and presence of multiple children infected in the household. Poor sanitation was defined as defecating in the fields or using superficial pits. Unsafe water intake was defined as not consuming water from the chlorinated municipal supply. Poverty likelihood based on the Simple Poverty Scorecard was not included in the regression model due to missing values in 17.6% of the sample. We used a p value < 0.2 for variable retention in the regression model.

The hot spots, cold spots, and spatial outliers analysis was performed in two different ways, one including all animals feces and another one including animals feces and household

infection status. Identification of hot and cold spots of *Fasciola* infected households was performed via Getis-Ord G statistics [18]. The statistic takes into account the interaction of the zone with itself plus the relationships with its surroundings and allows for a statistical significance evaluation of clustering via p-value as compared to a random distribution [19–20]. Other spatial patterns of the infection (outliers and clusters) in each household were analyzed by using Anselin Local Moran's I [18]. This index provides data regarding each location's risk based on statistically significant clustering patterns (high-high, low-low) and outliers (high-low, low-high) via a Z score [19]. The Z score indicates the presence of apparent similarity (clusters) or dissimilarity (outliers) as compared to that expected from random distribution. A positive Z score indicated clustering of high or low values (positive or negative cases) while a negative Z score indicated outliers. A high-low outlier represents a statistically significant high/positive value surrounded by low/negative values, while a low-high outlier represents the opposite. All maps were created under the WGS 1984 UTM Zone 18S coordinate system and Transverse Mercator as the elected projected system. A $p < 0.05$ was considered statistically significant.

## Results

Our study population consisted of 3000 children from 2122 households. After excluding 42 households lacking GPS information and 10 households where no livestock feces samples were collected, we included 2070 individual households in the analysis (Table 1). The median age was 9.1 years (6.7–11.8) and 1025 (49.5%) were female. Most children resided in the Anta district (50.8%). The median years of education was 6 (IQR 3–11) for the mothers and 9 (IQR 6–11) for the fathers. The mean poverty score was 39.6 (±10.7) indicating that about half of the study households had a 34.4% or higher likelihood of living under a USD 3.75/day poverty line. [16] The median elevation of the households was 3382 meters (IQR 3352–3479). Overall, 7.2% of the households had at least one *Fasciola* infected child. From the 611 households

**Table 1. Children demographic characteristics in *Fasciola* positive and negative households.**

|  |  | Total (N = 2070) | *Fasciola* Positive (n = 148) | *Fasciola* Negative (n = 1922) | p-value |
|---|---|---|---|---|---|
| | | Median (IQR)* | | | |
| Education of the father (years) | | 9 (6–11) | 7 (6–11) | 9 (6–11) | **0.007** |
| Education of the mother (years) | | 6 (3–11) | 6 (2–6.2) | 6 (4–11) | **<0.001** |
| Altitude of the house (meters) | | 3382 (3352–3479) | 3427 (3356–3610) | 3381 (3351–3477) | **<0.001** |
| | | Mean (+/- SD)* | | | |
| Poverty score | | 39.6 (±10.7) | 34.9 (±9.4) | 40.0 (±10.7) | **<0.001** |
| | | N (%)* | | | |
| Sex | Male | 1045 (50.5) | 74 (7.1) | 971 (92.9) | 0.902 |
| | Female | 1025 (49.5) | 74 (7.2) | 951 (92.8) | |
| Location | Anta | 1052 (50.8) | 71 (6.7) | 981 (93.3) | 0.085 |
| | Ancahuasi | 745 (35.9) | 64 (8.6) | 681 (91.4) | |
| | Zurite | 273 (13.1) | 13 (4.8) | 260 (95.2) | |
| Unsafe water consumption# | Yes | 90 (4.3) | 5 (5.6) | 85 (94.4) | 0.5484 |
| | No | 1980 (95.7) | 143 (7.2) | 1837 (92.8) | |
| Poor sanitation$ | Yes | 578 (27.9) | 56 (9.7) | 522 (90.3) | **0.0053** |
| | No | 1492 (72.1) | 92 (6.2) | 1400 (93.8) | |

[+] P value based on Mann-Whitney-U, $X^2$ as appropriate. Bolded values = significant to $p < 0.05$

*Median and IQR values provided if variables failed to have a normal distribution by Shapiro Wilk test

# Defined as main water source not coming from municipal supply

$ Defined as defecation in the opened field or in shallow pits.

where more than one child was tested for *Fasciola* infection, only 12 had multiple infected children. There were significant differences between fascioliasis positive and negative households in maternal and paternal years of education, poverty score, altitude of the household, and presence of unsafe sanitation (Table 1).

We collected 2648 samples of livestock feces, but only 2420 had GPS information and were included for distance analysis (Table 2). Cattle was the most frequent livestock species included (40.8%), followed by swine (32.8%), and sheep (26.4%). The overall *Fasciola* infection rate in livestock samples was 30.9%. A similar number of feces samples were collected from Ancahuasi and Anta with no significant differences in overall prevalence (Table 2). The highest infection prevalence was found in sheep feces (258/640, 40.3%) with Ancahuasi (129/292, 44.1%) as the place with the highest prevalence followed by Anta (109/290, 37.5%) and Zurite (20/58, 34.4%). Most cattle positive feces (328/987, 33.2%) were found in Anta (153/420, 36.4%) followed by Zurite (46/131, 35.1%) and Ancahuasi (129/436, 29.6%). Swine positive feces (163/793, 20.5%) were more often found in Ancahuasi (87/353, 24.6%) followed by Anta (66/363, 18.2%) and Zurite (10/77, 12.9%) (Table 2).

The median distance from a *Fasciola* positive household to the closest infected livestock feces sample was 44.6 meters (IQR = 14.7–112.8), while the median distance from a negative household to the closest infected livestock feces sample was 62.2 meters (IQR = 18.3–158.6) (p = 0.025) (Table 3). There was no significant difference between the distances from a *Fasciola* positive or negative household to a non-infected livestock feces sample (18.7 meters (IQR = 9.1–47.4) versus 22.4 meters (IQR = 9.1–66.5), p = 0.219). The median distance from a positive household to the closest infected cattle feces sample was 85.9 meters (IQR = 32–220.8) while the distance from a negative household to the closest infected cattle feces sample was 137.4 meters (IQR = 44.5–344.1) (p = 0.006). The distance of a positive household to infected swine feces (207.3 meters (IQR = 66.6–444.2)) was significantly different to the median distance from a negative household (236.9 meters (IQR = 89–610.5)) (p = 0.044). The distances from *Fasciola* positive or negative households to positive sheep feces samples were not significantly different. No differences were observed between *Fasciola* positive or negative households and negative feces samples from any livestock species (Table 3). The logistic regression models analyzing the effect of the distance to the closest animal feces on the household infection status adjusted by multiple infections in the household and altitude of the house demonstrated an OR 0.997 (95% CI 0.995–1.000, p = 0.0174). The effect modification on the model by distance to the closest feces by livestock species demonstrated no significant relationship to the household infection status (p = 0.7053). Similarly, the effect of the distance to the closest feces by animal status was not statistically significant (p = 0.9384).

The adjusted multivariable logistic regression using a 50-meter radius around households demonstrated a higher likelihood of finding cattle feces around infected households

**Table 2.** "Distribution of the livestock samples and comparison between *Fasciola* positive and negative samples ".

|  |  | Total (N = 2420) n (%) | Fasciola Positive (N = 749) n (%) | Fasciola Negative (N = 1671) n (%) | p-value* |
|---|---|---|---|---|---|
| Location* | Ancahuasi | 1081 (44.7) | 345 (31.9) | 736 (68.1) | < 0.535 |
|  | Anta | 1073 (44.3) | 328 (30.6) | 745 (69.4) |  |
|  | Zurite | 266 (11.0) | 76 (28.6) | 190 (71.4) |  |
| Species | Cattle | 987 (40.8) | 328 (33.2) | 659 (66.8) | **< 0.001** |
|  | Swine | 793 (32.8) | 163 (20.6) | 630 (79.4) |  |
|  | Sheep | 640 (26.4) | 258 (40.3) | 382 (59.7) |  |

*P value based on two-sided $X^2$ tests.

Bolded values = significant to p< 0.05

**Table 3. "Median distances from *Fasciola* positive or negative households to closest positive or negative livestock feces sample".**

|  | Positive Household (median, IQR) | Negative Household (median, IQR) | p-value [*] |
|---|---|---|---|
| Distances to the closest livestock feces sample (meters) |  |  |  |
| Positive livestock feces | 44.6 (14.7–112.8) | 62.2 (18.3–158.6) | **0.025** |
| Negative livestock feces | 18.7 (9.12–47.4) | 22.4 (9.1–66.5) | 0.219 |
| Distances to the closest feces sample according to livestock species (meters)[**] |  |  |  |
| Positive cow feces | 85.9 (32–220.8) | 137.4 (44.5–344.1) | **0.006** |
| Positive pig feces | 207.3 (66.6–444.2) | 236.9 (89–610.5) | **0.044** |
| Negative sheep feces | 82 (26.1–179.4) | 99.2 (33–199.1) | 0.235 |
| Negative pig feces | 49.1 (17–116.7) | 57.5 (19.6–147.6) | 0.290 |
| Negative cow feces | 48.7 (16.8–157.3) | 56 (18.2–165) | 0.578 |
| Positive sheep feces | 162.4 (82.7–317.2) | 147.9 (57.3–373.6) | 0.675 |

[*] P value calculated for Mann-Whitney-U statistics.

[**] Unable to report subgroup sizes, as feces might be close to more than one household.

Bolded values = significant to p< 0.05

independently of their infection status. However, infected households had a higher likelihood of being close to *Fasciola* infected cattle feces (OR = 1.89 (95%CI 1.31–2.73), p = 0.007) than to *Fasciola* negative cattle feces (OR = 1.42 (95%CI 1.06–1.89), P = 0.017). Using a 50-meter radius around households, *Fasciola* positive sheep feces were associated with a lower likelihood of the household being infected with *Fasciola* (OR = 0.52 (95%CI 0.32–0.84), p = 0.007). *Fasciola* negative sheep feces (p = 0.939) and swine feces regardless of their infection status (positive p = 0.7711, negative p = 0.7968) were not associated with infection in the households (Tables 4 and S1). Using a 100-meter radius in the adjusted logistic regression model, infected households still had a higher likelihood of having *Fasciola* positive cattle feces in their surroundings as compared to uninfected households (OR = 1.35 (95%CI 1.08–1.69), p = 0.049) but had a decreased likelihood of having *Fasciola* positive sheep feces (OR 0.77 (95%CI 0.60–0.99), p = 0.042). Any swine feces and *Fasciola* negative cattle or sheep feces showed no

**Table 4. "Multivariable logistic regression modeling factors assessing the likelihood of household positivity status by proximity to different types of livestock feces under different distance radius".**

| Radius length [*] | Odds Ratio | 95% Confidence Interval | p-value[***] |
|---|---|---|---|
| 50 meters [**] |  |  |  |
| Cattle negative | 1.42 | 1.07–1.89 | 0.017 |
| Sheep positive | 0.52 | 0.32–0.84 | 0.007 |
| Cattle positive | 1.89 | 1.31–2.73 | 0.001 |
| 100 meters [**] |  |  |  |
| Sheep positive | 0.77 | 0.60–0.99 | 0.049 |
| Cattle positive | 1.35 | 1.08–1.69 | 0.008 |
| 200 meters [**] |  |  |  |
| Cattle negative | 1.08 | 1.01–1.15 | 0.022 |

[*] The radius calculations included all animal feces, the house GPS was used in lieu of the feces sample GPS data when the information was missing.

[**] models adjusted by presence of poor sanitation, unsafe water, multiple household Fasciola infections, altitude of the household.

[***] Only statistically significant values to p< 0.05 are shown

association with household infection status (Tables 4 and S2). Using a 200-meter radius, infection in the household was only associated with *Fasciola* negative cattle feces in the adjusted analysis (OR = 1.08 (95%CI: 1.01–1.15), p = 0.022). Any swine or sheep feces, and *Fasciola* positive cattle feces showed no association with household infection status (Tables 4 and S3). All models were adjusted by presence of poor sanitation, unsafe water intake, and presence of multiple infectious within a household.

When assessing for effect of altitude in the household infection status, the models were analyzed by altitude strata at each radius using median altitude as the cutoff (S4 Table). Using a model adjusted for sanitation, safe water intake, and multiple household infections at 50, 100, and 200 meters radius, there were no variables associated with the household infection status in the lower altitude strata. At the higher altitude strata and using a 50 meter radius, the presence of cattle positive feces (OR = 2.22 (95% CI = 1.28–3.84), p = 0.0046), cattle negative feces (OR = 1.68 (95%CI = 1.16–2.43) p = 0.0058), sheep positive feces (OR = 0.59 (95%CI = 0.35–0.97), p = 0.0389) were associated with *Fasciola* positive household status. At the higher altitudes and using a 100 meters radius, cattle positive feces (OR = 1.51 (95%CI = 1.02–2.24), p = 0.0370), cattle negative feces (OR = 1.35 (95%CI = 1.07–1.70), p = 0.0100), and sheep positive feces (OR = 0.71 (95%CI = 0.52–0.99), p = 0.0446) remained associated with Fasciola infected households (S5 Table).

For the identification of outlier and cluster analysis of all livestock and household fascioliasis, we localized 76 high-high clusters, 726 low-low clusters, 62 high-low outliers, and 136 low-high outliers (Fig 2A). When only livestock feces were analyzed, we localized 141 high-high clusters, 338 low-low clusters, 65 high-low outliers, and 134 low-high outliers (Fig 2B). For the identification of hot/cold spots based on the confidence intervals, when livestock and household fascioliasis were considered, we identified 16 hot and 3 cold spots zones with over > 90% confidence (Fig 3A). We identified 4 hot and 3 cold spot zones when all livestock feces were considered at > 90% confidence (Fig 3B). When comparing data from households in hot and cold spots areas, hot spot households were located at higher median altitude (3574 meters (IQR = 3427.5–3764) versus 3473 meters (IQR = 3370–3497), p = 0.001), had lower socioeconomic scores (34 (IQR = 28.5–41) versus 43 (IQR = 36–49), p < 0.001), had less median years of education for the fathers (6 (IQR 5–10) versus 11 (IQR = 6–11), p < 0.001) and the mothers (4.5 (IQR = 2–6) versus 6 (IQR = 4–11), p < 0.001) than cold spot households.

## Discussion

Geographic coordinate systems and computational spatial analysis are important tools to study infectious diseases epidemiology [22]. Geographic information systems have been most useful for creating dynamic scenarios of infectious disease epidemics, forecasting transmission patterns of epidemic outbreaks, and assessing the effectiveness of infection control interventions [23]. Given its complex lifecycle, multiple reservoirs, patchy distribution, and emerging character, *Fasciola hepatica* is an ideal parasite to model using quantitative spatiotemporal analysis [24]. Using these tools, we found a positive association between environmental contamination with cattle feces and households where children with chronic fascioliasis reside. The presence of any cattle feces was associated with *Fasciola* infection in the household with increasing infection likelihood with decreasing distance. Cattle can pass between 60–106 pounds of manure daily which could explain a higher burden of environmental contamination compared to other livestock species [25]. Parkinson et al. found a significant correlation (R 0.769, p = 0.02) between human and cattle infection in 12 rural communities of the Bolivian Altiplano [26]. Mas Coma et al. in the same area, reported a high prevalence of cattle fascioliasis and, based on metacercaria production and infectivity, suggested that this livestock species

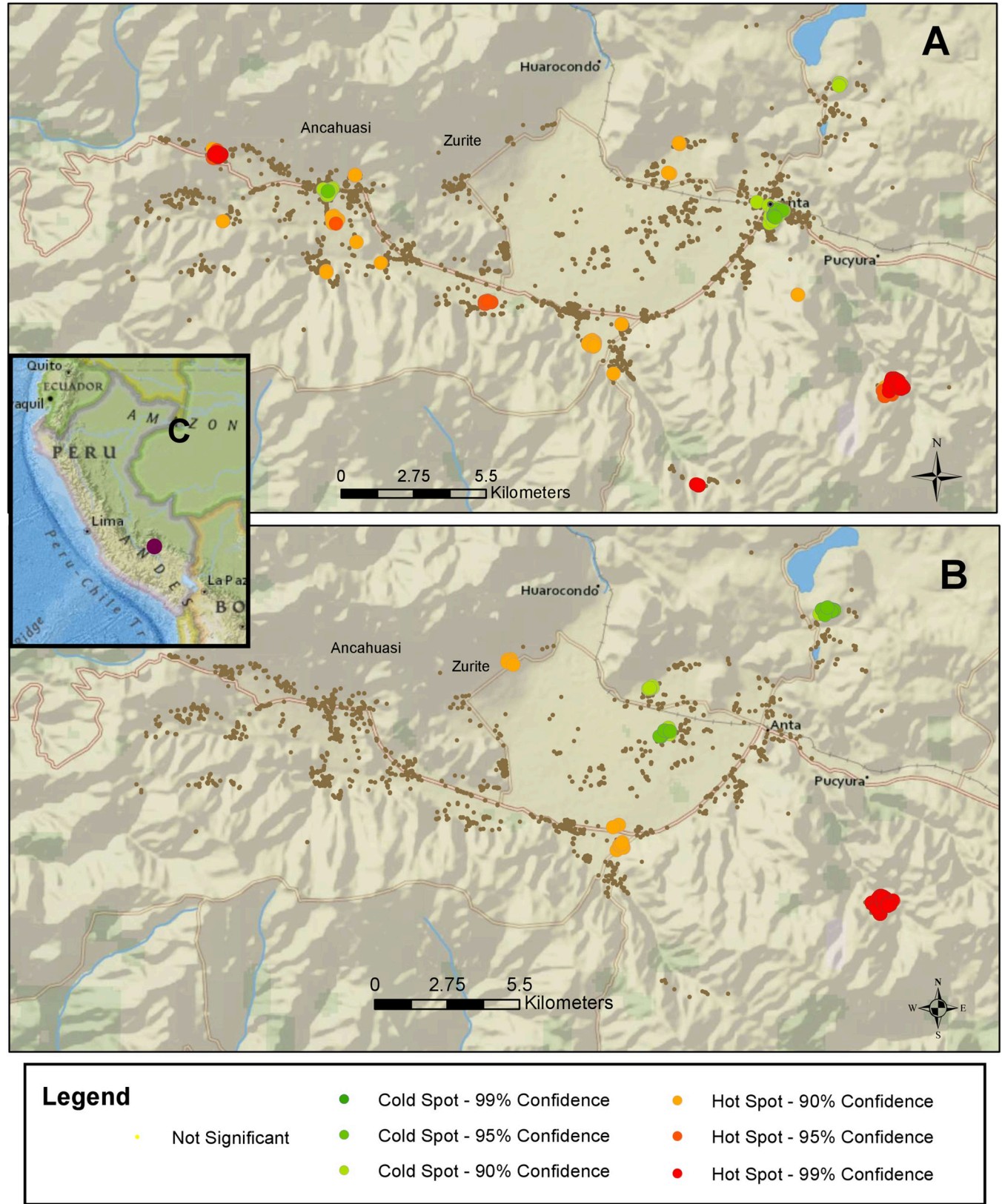

**Legend**

- Not Significant
- Cold Spot - 99% Confidence
- Cold Spot - 95% Confidence
- Cold Spot - 90% Confidence
- Hot Spot - 90% Confidence
- Hot Spot - 95% Confidence
- Hot Spot - 99% Confidence

**Fig 2. "Outlier and Cluster Analysis map based on Anselin Local Moran's I statistic of all livestock and human Fascioliasis in the Anta province of Cusco, Peru"** [21]. (A) Map based on infected and uninfected livestock and household infection status. (B) Map based on infected and uninfected livestock

feces (cattle, sheep, swine). (C) Inset showing the location of the study area (maroon dot) in the Cusco region of Peru. This map was created using ArcGIS v. 10.7.1 (ESRI, 2019). https://www.arcgis.com/home/item.html?id=67372ff42cd145319639a99152b15bc3.

plays an important role in maintaining transmission to human [27]. Other authors have suggested that consumption of contaminated water from natural sources or uncontrolled man-made canals is associated with fascioliasis transmission in humans and livestock [11,28]. Open defecation due to lack of latrines have been associated with Fasciola infection in observational studies in Bolivian Altiplano [11]. No direct transmission of fascioliasis between livestock and human is possible and the interactions between definitive hosts and snails are in need of further scrutiny to understand parasite transmission dynamics.

In our study, the presence of any cattle feces around the house, in particular *Fasciola* infected cattle feces, increased the likelihood of finding infection in the household and finding Fasciola infected sheep feces was associated with a decreased likelihood. These associations were maintained only at the higher altitude strata in the altitude stratified analysis. Higher altitude has been correlated with higher rates of fascioliasis likely due to increased survival of the intermediate host, longer cercarial shedding periods, and higher cercarial production [29]. Studies in South America have suggested an important role of sheep as *Fasciola* reservoirs at different altitudes based on infection prevalence [30]. Similarly, in our study, the prevalence of infection in sheep samples was the highest, but overall the proportion of sheep feces collected was the smallest compared to cattle and swine. It is unclear if this could have introduced sampling bias affecting our results. In the Anta province, sheep owners with the largest number of animals tend to keep herds close to pastures and away from their households especially at lower altitudes which may have also affected the evaluation of the environmental burden of sheep feces containing *Fasciola* eggs. Another potential explanation is a higher socioeconomic status among sheep owners which has been demonstrated to decrease the risk of infection in children [15]. The unexpected inverse association between household fascioliasis and sheep positive feces emphasize the complexities of *Fasciola* circulation between hosts and the need for further research on transmission.

Infected animal feces were significantly closer to infected households as compared to non-infected feces, especially cattle and swine feces. Experimental studies involving *Fasciola* eggs of sheep, cattle, and pigs have shown no differences in infection intensity or infectivity rates [30–31]. The particular association with cattle could be partially explained by the high number of this livestock species in the area probably constituting the main reservoir of fascioliasis in the Anta province and having a major role in transmission [11]. Using spatial analysis, we identified areas with high and low transmission of disease [32,33]. The large number of low-low outliers reflects the confidence on the locations of low risk of the disease. The small number of high-high outliers as compared to large number of "hot spots" when human data was added to livestock data, may suggest a discrepancy between human and animal risk of infection. In our study, the rates of human fascioliasis were considerably lower compared to the rates of animal fascioliasis. Areas of low-high and high-low transmission, remained similar when more data was added suggesting that in some locations of the Anta province infection risk is not well-defined [33]. In studies in Bolivia, the areas with higher prevalence of animal and human disease were located near water sources [10]. There is a lack of spatial analysis data about fascioliasis in South America and extrapolation of data from European studies might not be appropriate given differences in epidemiological behaviors. For example, spatial analysis done on Danish cattle herds showed that the presence of streams, wetlands, and pastures had a significant association with the presence of livestock infection [34]. While a study based in Switzerland, demonstrated that cattle were most likely infected in streams as they correspond to

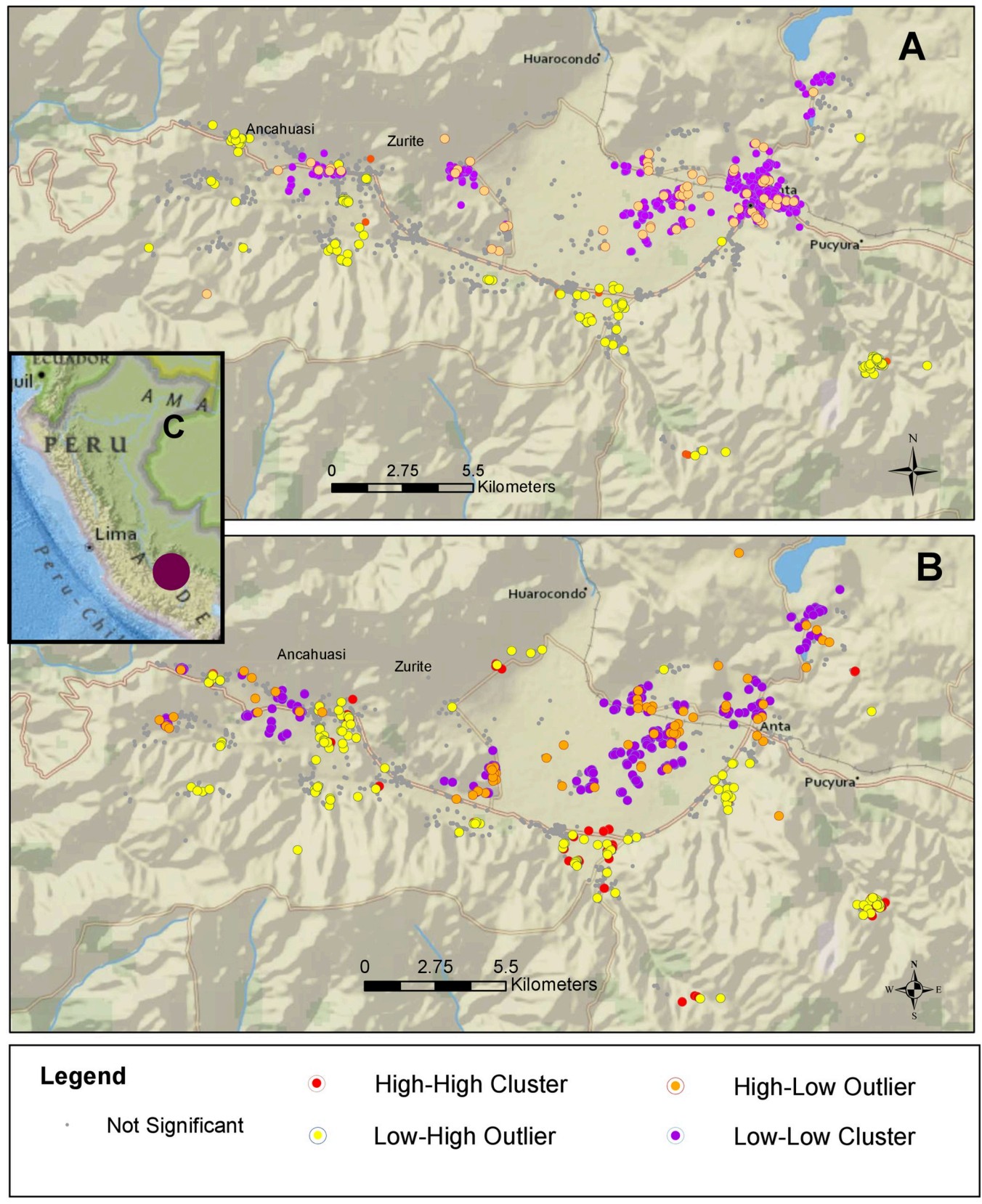

**Fig 3. "Significant hot/cold spot analysis of livestock and human fascioliasis in the Anta province of Cusco, Peru"** [18]. (A) Map based on infected and uninfected livestock feces and household infection status. (B) Map based on infected and uninfected livestock feces (cattle, sheep, swine). (C) Inset showing the location of the study area (maroon dot) in the Cusco region of Peru. This map was created using ArcGIS v. 10.7.1 (ESRI, 2019). https://www.arcgis.com/home/item.html?id=f33a34de3a294590ab48f246e99958c9.

areas where the snail prevalence is higher [35]. In this study, we did not account for distance to large water bodies, as this information was not available. In Figs 2B or 3B, we can locate a large water body (northeast of the border) where the small number of children sampled failed to show a higher infection rate. This could indicate a prominent presence of snails in smaller bodies of water influencing the infection prevalence in the area [36]. The hot spots were more likely to be in locations where poverty likelihood and altitude were higher and parent's education was lower compared to cold spots. Previous studies by our group, have associated lower maternal education status and lower socioeconomic status with increased *Fasciola* infection rates [15]. In the future, comparing the outliers with the differences in their corresponding environmental conditions could be analyzed to define other risk factors for fascioliasis.

There are several limitations in our study that need to be considered. We did not have data regarding distance to different waterbodies (small or large) which may have limited our analysis model. In our study, we were aiming to understand the rarely explored relationship between livestock and human fascioliasis. Livestock feces found around the households was collected and no samples were collected from individual animals. This precluded the calculation of fascioliasis prevalence in the different livestock species. Due to missing data, we were unable to use the socioeconomic score in the predictive model selection. However, elevation of the household correlated well with the socioeconomic score and we used it as a surrogate measure of poverty. Despite the relatively large sample of children tested for fascioliasis, the number of infected subjects was low limiting our statistical power. The lack of specific topographic variables for the predictive model could have decreased the accuracy of our risk estimates.

## Conclusion

In the Anta province of Cusco, the spatial distribution of *Fasciola hepatica* eggs in the environment was associated with the distribution of human fascioliasis. There was a different risk depending on livestock species with egg positive cattle and swine feces being independently associated with fascioliasis in the household. Further research to characterize areas where environmental egg contamination clusters with human cases is need to understand the dynamics of parasite transmission and identify potential strategies to optimize surveillance. The complexities of *Fasciola* infection among human and livestock emphasize the need for a one-health approach to research and control.

## Supporting information

**S1 Table. Number of livestock feces inside the 50 m buffer as per household status.** (DOCX)

**S2 Table. Number of livestock feces inside the 100 m buffer as per household status.** (DOCX)

**S3 Table. Number of livestock feces inside the 200 m buffer as per household status.** (DOCX)

**S4 Table. Adjusted univariate regression model for sheep positivity feces on household positivity status stratified by altitude of the household.** (DOCX)

**S5 Table. Adjusted Multivariate Logistic regression analysis stratified by higher or lower altitudes above the sea level associated with amount of livestock feces around different radiuses around the household.**
(DOCX)

## Acknowledgments

We thank the Cusco Regional Health Directorate of the Peruvian Ministry of Health. The contents of this manuscript are solely the responsibility of the authors and do not necessarily represent the official views of the National Institute for Allergy and Infectious Disease.

## Author Contributions

**Conceptualization:** Melinda Barbara Tanabe, John Prochaska, Miguel Mauricio Cabada.

**Data curation:** Melinda Barbara Tanabe, Maria Luisa Morales, Martha Lopez, Benicia Baca-Turpo, Eulogia Arque, Miguel Mauricio Cabada.

**Formal analysis:** Melinda Barbara Tanabe, John Prochaska.

**Funding acquisition:** Miguel Mauricio Cabada.

**Investigation:** Martha Lopez, Benicia Baca-Turpo, Eulogia Arque, Miguel Mauricio Cabada.

**Methodology:** Melinda Barbara Tanabe, Maria Luisa Morales, Miguel Mauricio Cabada.

**Project administration:** Maria Luisa Morales, Martha Lopez, Miguel Mauricio Cabada.

**Resources:** Martha Lopez.

**Software:** John Prochaska.

**Supervision:** John Prochaska, Maria Luisa Morales, Martha Lopez, Miguel Mauricio Cabada.

**Visualization:** Miguel Mauricio Cabada.

**Writing – original draft:** Melinda Barbara Tanabe.

**Writing – review & editing:** Melinda Barbara Tanabe, John Prochaska, Benicia Baca-Turpo, Eulogia Arque, Miguel Mauricio Cabada.

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
