## [Decision Letter · Decision Letter 0]

7 Feb 2022

Dear Dr. Cabada,

Thank you very much for submitting your manuscript "“Geospatial analysis of the associations between environmental contamination with Fasciola infected livestock feces and children in the Anta province of Cusco, Peru.”" for consideration at PLOS Neglected Tropical Diseases. As with all papers reviewed by the journal, your manuscript was reviewed by members of the editorial board and by several independent reviewers. In light of the reviews (below this email), we would like to invite the resubmission of a significantly-revised version that takes into account the reviewers' comments. 

We cannot make any decision about publication until we have seen the revised manuscript and your response to the reviewers' comments. Your revised manuscript is also likely to be sent to reviewers for further evaluation.

Sincerely,

Uwem Friday Ekpo, PhD

Associate Editor

Dileepa Ediriweera, PhD

Deputy Editor

Reviewer's Responses to Questions

**Key Review Criteria Required for Acceptance?**

**Methods**

-Are the objectives of the study clearly articulated with a clear testable hypothesis stated?

-Is the study design appropriate to address the stated objectives?

-Is the population clearly described and appropriate for the hypothesis being tested?

-Is the sample size sufficient to ensure adequate power to address the hypothesis being tested?

-Were correct statistical analysis used to support conclusions?

-Are there concerns about ethical or regulatory requirements being met?

Reviewer #1: Overall comments are that while the objective of the study was clear, the description of the study design was confusing and the statistical methods were under-described and inadequate for meeting the objectives. The sample size should be adequate and there are no obvious issues with ethical requirements.

Specific comments on methods: 

Feces and stool are both used in this paper. Pick one and be consistent in use.

Lines 95-96. More detail is needed on how the three fecal samples were collected, especially from older children, and when this occurred in the context of a cross-sectional household visit. Were there repeat visits?

Line 98. My understanding is that children were enrolled, then had 3 stools collected and tested. However, line 98 suggests that this study was instead a two-staged process with pre-testing of lots of kids to identify “households with multiple children infected with Fasciola”, followed by some sort of secondary enrollment of “only one infected child selected to participate while other children in the household with or without the infection were excluded.” By default if you are testing child stool for Fasciola, then they are participating. Please revise to improve clarity.

Lines 100-102. Are these animals of the participating household or potentially those of a neighbor? Also, were any criteria used to eliminate animal stool that seemed old or desiccated?

Line 108. “Children’s” or “Child”

Lines 113-114. Related to my comment about lack of clarity on study enrollment, the difference between “Chronic fascioliasis” and “Positive Fasciola household” outcomes is not clear from the study design, and the individual-level fascioliasis outcome is not reported or analyzed as an outcome anywhere in the analysis, although it should be. 

Lines 124-131. Why are you using backward regression modeling with only four (three in practice) variables included in the model? If this was truly the process, then you need to provide the full stepwise rules you used for evaluating model fit at each stage of model simplification. Related to this and my comment above, why has the model not also been adjusted for multi-infections among household members and proximity of water bodies with snails? Children could be infected by siblings shedding eggs into the environment as easily as from animals. And, as you clearly stated in the introduction, none of this matters without the intermediate snail host. The model is poorly examining the obligate components of the transmission cycle (Figure 1).

Lines 128-129. The count variable is not reported anywhere in the results. If feces counts for each species are used to code species infection status variables also included in the model, these needs to be described.

Lines 131-136. This is not an adequate description of the cluster analysis intent and method. What specific relationships are you probing to define as hot zone or cold zone? Jumping ahead to Figure 2, it appears your research questions include animal to animal geospatial relationships and animal to household. This all needs to be described.

Reviewer #2: Needs a little more on snail habitats

Reviewer #3: Objectives are clear.

Design not appropriate.

Population appropriate.

Sample size almost sufficiient.

Statistical analyses correct.

Ethical approvals need more data.

**Results**

-Does the analysis presented match the analysis plan?

-Are the results clearly and completely presented?

-Are the figures (Tables, Images) of sufficient quality for clarity?

Reviewer #1: The analysis do aim to address the study objective but do not match the analysis plan as described in methods, the results lack detail, and Tables and Figures lack organization and sufficient detail to be free standing summaries of presented material.

Specific comments on results:

Line 146-147. Please also report how this sample size was achieved in terms of number of children per year, whether there were year-to-year differences in prevalence, and how often you came across multiple children testing positive in the same household. Also, “households” should be singular.

Table 1. Age is reported as mean (SD) while other quantitative variables with a potentially normally distributed distribution like altitude and poverty score are in median (IQR). Did these two variables in particular not follow a Gaussian distribution?

Lines 161-165. Were there any differences in types of animals by geographic district that might occur for example by clusters of livestock production sectors?

Line 165-166. The statement “Most feces samples were collected in the Ancahuasi district and more samples were positive in this district” does not accurately reflect what is shown in Table 2. There are extremely minimal differences in number of samples collected and positivity rate between Ancahuasi and Anta.

Table 2. There is an “n” missing in “Total (N = 2420)(%)”. The Table header could be made clearer by putting either the (total N) or the n (%) on its own row.

Lines 170-182 and Table 3. I believe this analysis is missing a final step for interpretability. Individual tests for differences in distance by positive or negative animal status and household outcome is confusing in presentation and resulting in non-interpretable significant associations. Closer scrutiny of the distances shows that households with a fasciola positive child are more likely to be close to cattle and swine feces, regardless of whether the animal feces itself was positive for the parasite, with sheep going in the opposite direction. The lack of statistical association in the animal-negative comparisons is mostly likely limited power to detect differences due to small sample sizes in the household-positive group, although the lack of reporting on group size makes this difficult to tell. A better way to do this would be to estimate the effect of distance to the animal feces sample on household positivity status, then test for effect modification by animal stool sample testing status. If the relationship between distance and outcome is modified by animal stool testing status, then report sub-group effects. Or, calculate the sub-group conditional probabilities for increases or decreases in distance on household status and the risk ratio for the probability of positive household status where the animal stool contains parasites versus where it does not. For cattle and swine, you will likely see no impact of animal feces positivity on the distance to outcome relationship, although sheep might be interesting. Be sure to adjust these models for altitude and (see above) presence of other positive children in household and water bodies.

Table 3. Report your sub-group sizes and be consistent in presenting data. Either always do negative then positive feces or vice a versa. Similarly, your species specific positives and negatives are not organized in any specific way, much less aligned in flow with the way material is presented in the results text.

Line 195. Should be “Table 4”.

Line 189-204 and Table 4. I like the idea of this spatially comprehensive analysis approach more than the oversimplified analysis of median distance with one animal feces sample, as commented on above. However, you have not yet demonstrated the value of this multi-animal stool assessment approach by describing overall count or proportion of positive animal stools out of total counted within 50, 100, or 200 for positive and negative households. Another approach if your total counts are low is to use ratio of positive to negative at <50 meters, and <100 meters, and <200 meters. This could be put in a table in supplemental materials, but is necessary for defending exposure variable validity and for explaining why you simplified what you said you modeled (“overall count of animal feces” and “the individual species sample infection status”) vs what you report (sheep positive, cattle positive, swine positive). For that matter, I cannot figure out how the effects for the variables reported relate to the variables included in the model. A second point on the analysis method is that on lines 124-126 you stated you were using a backward selection modeling process. Backwards selection suggests you were seeking model parsimony, rather than fully adjusted effects. If that is the case, you must describe whether variables were removed, in what order, and how much that improved model fit. Considering you were only starting with the 2 variables above and altitude, I am not clear what the purpose for backwards selection was.

Table 4. (1) Please reorder the types of animals so that they are consistent across each distance unit, either Swine, Sheep, then Cattle, or Sheep, Swine, then Cattle. The 200 meter comparison lacks Sheep positivity altogether with no explanation. (2) The title is not descriptive of the analysis, which assessed the likelihood or odds of household positivity status by proximity to different types of domesticated animal vectors. (3) The table must be free standing, meaning when you say “multivariable”, you need to clarify in the footnote what variables were included in the model, either as exposure variables or interest or confounders. 

Table 4. Please clarify what you mean by using household GPS where animal stool gps data was missing and how this relates to the (more statistically appropriate) statement on line 122 that “Livestock feces without GPS data were excluded from the buffer/radius analysis”.

Lines 208-214 and Figure 2. Define high-high, low-low, etc. in methods and Figure footnotes, and be sure to provide some context to help readers interpret what clusters of these mean in the figures. Also, you cannot cite Figure 2B before 2A; either flip your pictures or your text so 2a comes first.

Reviewer #2: Yes

Reviewer #3: The analysis presented matches the analysis plan.

Results do not consider all factors.

Figures to be pronouncedly improved.

Tables need clarifications.

**Conclusions**

-Are the conclusions supported by the data presented?

-Are the limitations of analysis clearly described?

-Do the authors discuss how these data can be helpful to advance our understanding of the topic under study?

-Is public health relevance addressed?

Reviewer #1: I do not believe the results, as they exist currently, support the conclusions. The limitations are not complete. Finally, the impact of the results on broader understanding is limited by not considering the role of within-household transmission and the conditional dependency of transmission on intermediate snail vectors in the explicit study design and analysis plan.

Lines 240-243. Per my comments above about weaknesses in the analysis approach, this conclusion is not yet supported by the evidence.

Lines 279-281. Where is this information? Any sort of assessment about location of water bodies that could harbor the snail vectors would address my comment about this being lacking as an adjustment factor in the analysis.

Lines 281-284. Please explain contextual situation in Cusco further to explain how poverty in safe drinking water and sanitation would facilitate completion of the Fasciola life cycle.

Lines 288-295. The limitation should also acknowledge that analysis did not account for the role of other household members as influences on the index child’s infection status, nor the dependency of any human to human or animal to human transmission on availability of intermediate snail vectors. Directionality cannot be inferred from this cross sectional study design, and it is also possible that children could be sources of infections in the animals in their community. Adjusting for household conditions like use of latrines/toilets would have improved such inference.

Reviewer #2: (No Response)

Reviewer #3: Conclusions pose doubts because of methodology and overlook of factors.

Limitations section to be widely extended.

Authors discussion needs improvements.

Public health relevance is addressed.

**Editorial and Data Presentation Modifications?**

Reviewer #1: These recommendations are embedded in the section by section review, but generally, the tables lacked organization and both tables and Figures needed footnotes and improvements in titles.

Reviewer #2: (No Response)

Reviewer #3: See below

**Summary and General Comments**

Reviewer #1: A geospatial analysis of proximity to animal feces with parasites on human infection is valuable. The biggest issue is that while authors used rigor in confirming Fasciola infection in a large subset of children and local animals, they did not apply such rigor to measuring and analyzing all required conditions for Fasciola transmission. This lack in comprehensive analysis was surprising given the acknowledgement of the complex life cycle in the introduction and Figure 1. The dependency of Fasciola on a intermediate host for maturation and infectivity means that all relationships between infected and susceptible hosts is by default conditional on the host presence, and therefore the mechanism underlying and validating any relationships between animal feces and households with infections is unclear. A vague reference in discussion to location of water bodies suggests this ecological data might be available to authors but the manuscript as it stands does not suggest proof of mechanistic relationships can be resolved. Another major issue is that while multiple within-household infections was considered at some point in study design, it was not considered important as a confounder in the detection of Fasciola among siblings and nearby domestic animals and when and why multiple children in the household were tested is unclear. This may be a point that authors can easily address through adjustment of the analysis approach and would improve the interpretability of the results. Last, the methodological section suggests authors could greatly improve analysis by seeking council from a biostatistician or epidemiologist.

Reviewer #2: (No Response)

Reviewer #3: This manuscript has the purpose of evaluating the spatial relationships between fascioliasis infection in children and the environmental burden of Fasciola eggs passed by different livestock in the Cusco region of Peru, that is the presence of liver fluke eggs on fecal samples collected on the soil.

The manuscript includes many methodological problems, the baseline knowledge on the scenario is clearly insufficient regarding key factors, and there are several misinterpretations of basic concepts of the disease. Nevertheless, the originality of the approach merits an effort in the way for the needed improvement.

1.- The first general consideration refers to the literature used. One may conclude that authors consider that human fascioliasis follows similar transmission and epidemiological characteristics everywhere. This disease is markedly heterogeneous in epidemiological facies. So, using examples from Nepal, Pakistan, Egypt, Iran, Ghana, Zambia, Indonesia, Denmark, Switzerland, or Malaysia, has no sense here. Fascioliasis in Andean areas differs very pronouncedly from fascioliasis in the aforementioned countries. So, the literature should be reviewed and articles referring to fascioliasis in maximum South America used, above all concerning altitude areas in Andean zones.

2.- The second problem concerns the lack of knowledge, or at least the overlooking of key epidemiological aspects, regarding the study area:

A) The origin of human infection is mainly through contaminated vegetables, predominantly freshwater plants, but also drinking of natural water or beverages made with sylvatic plants and natural water. In all the human endemic areas in the Andes, the infection of children occurs mainly (i) along the way to and from the school, (ii) when accompanying livestock for grazing and drinking in freshwater collections, and (iii) at home when eating contaminated plants and drinking natural water collected outside. This means that taking the household as the reference for such an analysis is not appropriate, i.e. for instance the distance from the school may be more important.

B) The dependence of the rural inhabitants from natural water sources lead them to choose places for their dwelling located close to freshwater collections. These freshwater collections used to be inhabited by lymnaeid vectors and to be visited by livestock for drinking. So, these foci act for the infection of both humans and animals.

C) In these rural areas, livestock moves freely around. Fences are not used, so that cattle, sheep and pigs use to appear mixed in the same grazing area. And the image one gets from a given place may differ one day from another, and one season from another Thus, no sense to compare with countries where animals are strictly controlled in farms, livestock maintained within fences, and separated according to species.

D) Rural inhabitants give, however, more importance to cattle because of milk, meat and even family prestige inside the community. This means that the type of ownership by each family is a crucial factor to be considered. The infection risk in a household of a big owner may be different from that of a small owner.

The following two papers may help in showing the aforementioned aspects of disease heterogeneity in humans worldwide and its complexity in the neighboring Bolivia:

Angles R. et al., 2022. One Health action against human fascioliasis in the Bolivian Altiplano: Food, water, housing, behavioural traditions, social aspects, and livestock management linked to disease transmission and infection sources. Int. J. Environ. Res. Public Health 19, 1120, 44 pp.

Mas-Coma S. et al., 2018. Human fascioliasis infection sources, their diversity, incidence factors, analytical methods and prevention measures. Parasitology 145 (13, Special Issue): 1665-1699.

3.- Lines 47-48: "The factors associated with transmission of the disease among humans have not been well described." This is no true. Add "... in the area studied." or similar.

4.- Lines 74-77: "However, studies in areas of the world with lower prevalence of infection have reached different conclusions.[11] A study in Qena, Egypt showed an animal prevalence ranging from 17.2 to 33.7%, but no evidence of human fascioliasis in the same region.[12]." This has no sense here. Just for information: human fascioliasis concentrates in Lower Egypt, that is, in the widely irrigated Nile Delta, where G. truncatula with other lymnaeid vectors are present, whereas Qena is in Upper Egypt, surrounded by desert and with only R. natalensis present. In the Nile Delta, human infection intensities similar to those in the Andes have recently been found. See the following:

Periago M.V. et al., 2021. Very high fascioliasis intensities in schoolchildren of Nile Delta governorates: The Old World highest burdens found in lowlands. Pathogens, 10: 1210, 20 pp.

5.- Line 78: Differences of results between studies are not inconsistencies, but the reflection of different transmission patterns and epidemiological situations. There are many articles on that.

6.- Figure 1: This drawing cannot be accepted and reflects a lack of expertise on fascioliasis. This should be corrected, if not there are afterwards students reproducing it in their master's theses, etc. Please correct the following: (i) eggs do not seek an intermediate host, this is for the hatched miracidium to do; (ii) the drawing of the snail represents a terrestrial snail, with eyes on the tips of the tentacles and the shell of an helicid or planorbid; when one sees a lymnaeid, one never forgets it; (iii) illustrating a cercaria by means of a drop does not appear to be logical; please substitute the drops by simplified cercariae with their body and long tail; (iv) cercariae do not infest aquatic plants, it is up to the encysted metacercariae to attach to the vegetables.

7.- Line 114: How and where the livestock feces were collected should be specified in detail. Simply collecting them from the soil? Cattle feces are well visible, but feces from sheep and pigs may be easily overlooked. This may underlie an important bias in the results, because of inadvertently giving more weight to cattle. How was this solved?

8.- Line 115: Which quality control procedure was used?

9.- Line 127: How were areas around each household at 50, 100, and 200-meter radius delineated. In the map with Google Earth? If in the field, how were the 200 m measured?

10.- Line 137-138: The numbers and dates of the official approval documents should be added.

11.- Table 1: It needs clarifications to be added to the legend. For instance, parentheses sometimes seem to include means, but in other cases they refer to ranges.

12.- Table 3: Something does no fit well here. Positive feces in sheep and pig are clearly higher than positive feces in cattle. This contradicts the conclusion of the manuscript.

13.- Figure 2: There is no way to know which color of the circles concern cattle, sheep and pigs. Subfigure C has no sufficient resolution.

14.- Figure 3: Similar interpretation problem as in Figure 2.

15.- Lines 243-245: More manure per cow does not mean a higher number of liver fluke eggs, but eggs more diluted.

16.- Line 251: Add "in the Cuzco area" or similar. In the Bolivian Altiplano, all these aspects are already well elucidated.

17.- Lines 264-265: "Infected animal feces were significantly closer to infected households as compared to non infected stool, especially cattle and swine feces." This means that dwellings are located close to freshwater collections, as in the Altiplano.

18.- Lines 276-279: Delete these examples about Denmark and Switzerland. Situations are not comparable.

19.- Lines 282-284: Again, for the same reason, delete the example of Malaysia, which moreover deals on intestinal monoxenous helminths. Same with the example of Iran. Use the aforementioned reference of Angles et al. on the Altiplano to refer to the link with the socioeconomic status. There is also an article on that aspect in Cajamarca - see in references of Angles et al.

20.- Lines 288-295: This section is a good place to again refer to what has been noted above in points 2 A-D, although these crucial aspects should be referred to at least in the Discussion.

21.- Lines 297-298: Add "... eggs shed by cattle, sheep and goats" to avoid confusion with human stools. Outdoor defecation by rural inhabitants is commonly practiced in the Andean altitude areas.

22.- Line 301: Change "need" to "needed".

23.- Line 302: Change "surveillance" to "control measures".

PLOS authors have the option to publish the peer review history of their article (what does this mean?). If published, this will include your full peer review and any attached files.

Reviewer #1: No

Reviewer #2: No

Reviewer #3: No
---

## [Decision Letter · Decision Letter 1]

14 May 2022

Dear Dr. Cabada,

We are pleased to inform you that your manuscript '“Geospatial analysis of the associations between environmental contamination with Fasciola infected livestock feces and children in the Anta province of Cusco, Peru.”' has been provisionally accepted for publication in PLOS Neglected Tropical Diseases.

Best regards,

Uwem Friday Ekpo, PhD

Associate Editor

Dileepa Ediriweera

Deputy Editor

Reviewer's Responses to Questions

**Key Review Criteria Required for Acceptance?**

**Methods**

-Are the objectives of the study clearly articulated with a clear testable hypothesis stated?

-Is the study design appropriate to address the stated objectives?

-Is the population clearly described and appropriate for the hypothesis being tested?

-Is the sample size sufficient to ensure adequate power to address the hypothesis being tested?

-Were correct statistical analysis used to support conclusions?

-Are there concerns about ethical or regulatory requirements being met?

Reviewer #2: (No Response)

Reviewer #3: All Ok

**Results**

-Does the analysis presented match the analysis plan?

-Are the results clearly and completely presented?

-Are the figures (Tables, Images) of sufficient quality for clarity?

Reviewer #2: (No Response)

Reviewer #3: All Ok

**Conclusions**

-Are the conclusions supported by the data presented?

-Are the limitations of analysis clearly described?

-Do the authors discuss how these data can be helpful to advance our understanding of the topic under study?

-Is public health relevance addressed?

Reviewer #2: (No Response)

Reviewer #3: All Ok

**Editorial and Data Presentation Modifications?**

Reviewer #2: (No Response)

Reviewer #3: All Ok

**Summary and General Comments**

Reviewer #2: Revision is satisfactory

Reviewer #3: This new version has clarified the problems raised by this reviewer and the manuscript has markedly improved its contents.

Just to help a little bit more with additional information (not for manuscript modification):

1. Lines 85-87: Leave the text as it is if you want, but, just for your information, the reason for the differences inside Brazil have already been elucidated molecularly, experimentally and in the field. Indeed, from the point of view of fascioliasis Brazil appears to be a continuation of Uruguay because of historical reasons. Unfortunately Schwantes et al. 2019 overlooked the corresponding study were all this is clarified. The main factors causing the differences are altitude and snail vector species. See the following:

Bargues et al., 2017. DNA multigene characterization of Fasciola hepatica and Lymnaea neotropica and its fascioliasis transmission capacity in Uruguay, with historical correlation, human report review and infection risk analysis. PLoS Neglected Tropical Diseases, 11 (2): e0005352 (33 pp.).

2. Discussion, line 363: Just for your information, lymnaeids at altitude are found in small water bodies, not in large ones. See the following:

Bargues et al., 2021. One Health initiative in the Bolivian Altiplano human fascioliasis hyperendemic area: Lymnaeid biology, population dynamics, microecology and climatic factor influences. Brazilian Journal of Veterinary Parasitology, 30 (2): e025620 (24 pp.).

3. Discussion, line 366-368: For your information, the significant direct relationship between altitude and prevalences in children has already been established in Andean valleys where an altitudinal gradient can be assessed. See the following:

Gonzalez et al., 2011. Hyperendemic human fascioliasis in Andean valleys: An altitudinal transect analysis in children of Cajamarca province, Peru. Acta Tropica, 120: 119-129.

PLOS authors have the option to publish the peer review history of their article (what does this mean?). If published, this will include your full peer review and any attached files.

Reviewer #2: No

Reviewer #3: No

---

## [Editor Report · Acceptance letter]

13 Jun 2022

Dear Dr. Cabada,

We are delighted to inform you that your manuscript, "“Geospatial analysis of the associations between environmental contamination with Fasciola infected livestock feces and children in the Anta province of Cusco, Peru.”," has been formally accepted for publication in PLOS Neglected Tropical Diseases.

Best regards,

Shaden Kamhawi

co-Editor-in-Chief

Paul Brindley

co-Editor-in-Chief
